# A Multi-Task Benchmark for Abusive Language Detection in Low-Resource Settings

**Fitsum Gaim**      **Hoyun Song**      **Huije Lee**
**Changgeon Ko**      **Eui Jun Hwang**      **Jong C. Park**

Korea Advanced Institute of Science and Technology (KAIST)
{fitsum.gaim,hysong,huijelee,pencaty,ehwa20,jongpark}@kaist.ac.kr

## Abstract

Content moderation research has recently made significant advances, but remains limited in serving the majority of the world's languages due to the lack of resources, leaving millions of vulnerable users to online hostility. This work presents a large-scale human-annotated multi-task benchmark dataset for abusive language detection in Tigrinya social media with joint annotations for three tasks: abusiveness, sentiment, and topic classification. The dataset comprises 13,717 YouTube comments annotated by nine native speakers, collected from 7,373 videos with a total of over 1.2 billion views across 51 channels. We developed an iterative term clustering approach for effective data selection. Recognizing that around 64% of Tigrinya social media content uses Romanized transliterations rather than native Ge'ez script, our dataset accommodates both writing systems to reflect actual language use. We establish strong baselines across the tasks in the benchmark, while leaving significant challenges for future contributions. Our experiments demonstrate that small fine-tuned models outperform prompted frontier large language models (LLMs) in the low-resource setting, achieving 86.67% F1 in abusiveness detection (7+ points over best LLM), and maintain stronger performance in all other tasks. The benchmark is made public to promote research on online safety.[1]

## 1  Introduction

The proliferation of social media has revolutionized global communication, enabling unprecedented connectivity while simultaneously creating new vectors for harm through abusive content [1]. Online hostility and harassment affect millions of users, particularly vulnerable groups including minors and minority communities, often causing physical and psychological harm while reinforcing social marginalization [2, 3]. Although significant progress has been made in automated detection of abusive content for high-resource languages such as English [4–6], the majority of the world's low-resourced languages, such as those spoken in Africa, remain understudied [7], creating an alarming disparity in online safety and protection.

Tigrinya, a language with approximately 10 million speakers mainly in Eritrea and Ethiopia, exemplifies this technological divide [8]. Despite its significant speaker population, Tigrinya remains computationally under-resourced with minimal datasets, tools, and models [9]. In particular, there is a lack of well-established benchmarks to gauge progress in content moderation research. This gap exposes the Tigrinya-speaking communities to unchecked online abuse. More broadly, it highlights

---

[1] TiALD Resources (Dataset, Code, Models): https://github.com/fgaim/TiALD

39th Conference on Neural Information Processing Systems (NeurIPS 2025) Track on Datasets and Benchmarks.

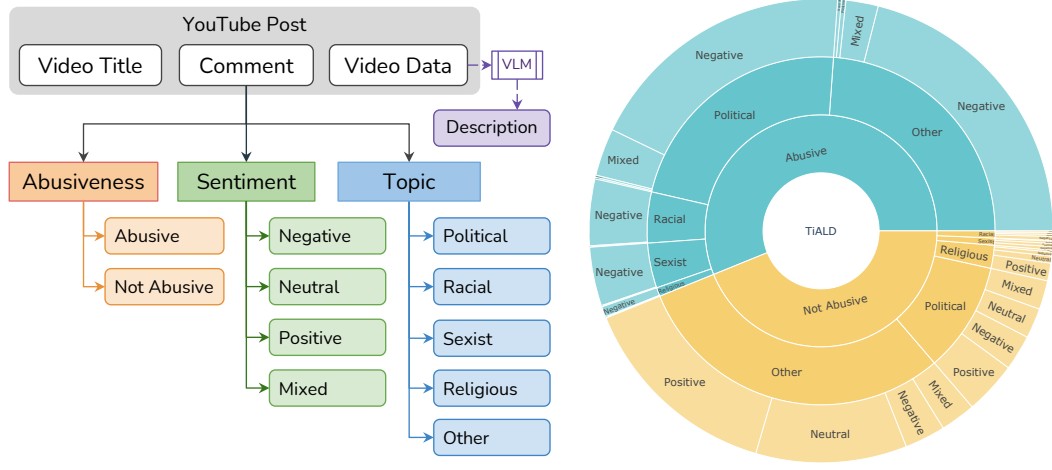

(a) Dataset Schema and Features      (b) Task-wise Class Distributions

Figure 1: An overview of the TiALD Dataset Design and Annotated Class Distributions.

the urgent need for dedicated computational resources and methods of content moderation research in underrepresented languages.[2]

In this paper, we address the critical gap by introducing the **Ti**grinya **A**busive **L**anguage **D**etection (**TiALD**) dataset, a large-scale human-annotated multi-task benchmark for abusive language detection in the Tigrinya language. TiALD adopts a multifaceted approach, providing joint annotations for three tasks: abusiveness detection, sentiment analysis, and topic classification across 13,717 manually annotated YouTube comments. We further enrich the dataset by generating descriptions of the visual content of the corresponding videos using a Vision-Language Model (VLM), enabling the relational analysis against the user comments. Our annotation scheme enables richer contextual understanding of abusive content, supporting more nuanced analysis than that with the typical binary classification setups alone. Figure 1 shows an overview of the dataset design and class distributions. The contributions of this work can be summarized as follows:

- We present the first large-scale multi-task benchmark dataset for abusive language detection in Tigrinya based on user comments collected from YouTube channels in the community.
- We propose a data selection methodology for iterative semantic clustering of terms to address the inherent imbalance in abusive vs. non-abusive content on social media.
- We accommodate the sociolinguistic reality of Tigrinya social media by covering posts written in both the standard Ge'ez script and Latin transliterations, ensuring that the trained models handle actual language practices.
- We demonstrate that small, specialized models with joint multi-task learning of abusiveness, topic, and sentiment tasks outperform large frontier models, establishing a strong baseline.

## 2 Dataset Construction

In this section, we describe the construction of the TiALD dataset and an analysis of the annotations.

### 2.1 Data Collection and Preprocessing

As the source of data, we initially collected 4.1 million comments from 51 popular YouTube channels with more than 34.5K videos and a total of over 2.2 billion views at the time of collection. These channels cover various genres, including news, entertainment, music, educational, documentaries,

---

[2] Monitoring and Analyzing online content is a major commitment of the United Nations' *Strategy and Action Plan on Hate Speech* established in 2019: `https://www.un.org/en/genocideprevention/documents/advising-and-mobilizing/Action_plan_on_hate_speech_EN.pdf` (accessed on 2025-04-19).

vlogs, and more, ensuring a diverse and representative sample of the Tigrinya-speaking social media landscape. We then preprocessed the data by filtering out non-text comments, such as those that contain only emojis and also non-Tigrinya comments, such as those written fully in English, Arabic, Amharic, etc, using the GeezSwitch library [9]. Within the Tigrinya content, we observed that around 64% of the collected source comments contained Romanized text, where users employ improvised and non-standard transliteration schemes. To accommodate this, we cover comments written in both scripts in the annotation process, 70% Ge'ez and 30% Latin or mixed, which could help develop models that reflect a realistic usage of the language in social media.

## 2.2 Data Samples Selection for Annotation

Abusive language constitutes a minority of online content, making random sampling inefficient for dataset construction, while simple keyword searches yield lexically homogeneous datasets. Moreover, most low-resourced languages lack extensive curations of abusive terms for this purpose. Recognizing this gap, we propose a semi-automatic strategy that leverages a vector space of candidate samples and a small set of seed terms to effectively expand the selection criteria without heavily relying on the lexical search of manually curated terms. To this end, we trained word embeddings on the original 4.1 million comments from YouTube using word2vec [10] implemented in Gensim [11] with the CBOW architecture and a vector dimension of 300. We used cosine similarity to compute nearest neighbors and then applied an iterative process of term expansion and deduplication to construct a diverse and balanced annotation pool.

**Seed Word Selection and Iterative Expansion.**   As a first step, we curated an initial set of seed words representing the target classes: abusive, non-abusive, political, religious, etc. These seed words included not only derogatory terms but also common words, political party names, ethnic groups, and religious terms, totaling 61 terms across all categories. We then expanded this seed set through a three-stage iterative search in the embedding space, designed to maximize lexical diversity while maintaining semantic relevance. In the first stage, for each seed term $w_0$, we retrieved its 50 nearest neighbors, retaining only those that are morphologically distinct from $w_0$ (i.e., not simple inflections). In the second stage, for each term obtained in Stage 1, we retrieved 25 additional nearest neighbors, filtering out simple derivations from the source terms. In the third stage, we retrieved 10 nearest neighbors for each term from Stage 2, applying the same distinctness criteria.

This iterative expansion process yielded 8,728 diverse and representative terms. We then selected 15K comments covering the expanded term list, with each term appearing in at least two comments, ensuring coverage across all categories. To this set, we added 5K randomly sampled comments from the remaining corpus as a control group. Our approach achieved a substantially higher type-to-token ratio of $0.28$ compared to $0.13$ for pure random sampling, resulting in a balanced pool of 20K comments ready for human annotation.

## 2.3 Data Annotation

We hired nine native speakers as annotators, four females and five males, between the ages of 22 to 48 years. The annotators were asked to label each comment for three tasks: abusiveness, sentiment, and topic. For **Abusiveness**, comments are categorized into two classes (abusive or not abusive), providing the primary classification target. The **Sentiment** dimension adds emotional context with four possible classifications (positive, neutral, negative, or mixed). Finally, the **Topic** classification assigns each comment to one of five categories (political, racial, sexist, religious, or miscellaneous topics), capturing the subject matter of the context. The annotation schema of the three tasks is depicted in Figure 1a. The annotation campaign was conducted in a controlled setting with informed consent from each participant, and a set of instructions and annotation guidelines was provided to ensure the quality and consistency of the dataset. By the end of the process, we collected annotations for 13,717 comments, and we measured the inter-annotator agreement as discussed in Section 2.4.

**Generating Video Descriptions.**   To provide richer contextual information for analysis, we extended the dataset with descriptions of the visual content in the videos corresponding to comments in the evaluation splits. These descriptions enable researchers to investigate potential relationships between video content and abusive language, such as whether certain visual elements might trigger hostile comments. We generated these descriptions using the Qwen-2.5-VL 3B [12] and refined them with

Table 1: TiALD Dataset: Distribution of the Three Tasks and Dataset Splits.

| Task | Label | Train | Test | Dev | Samples |
|---|---|---|---|---|---|
| Abusiveness | Abusive | 6,980 | 450 | 250 | 7,680 |
| | Not Abusive | 5,337 | 450 | 250 | 6,037 |
| Sentiment | Positive | 2,433 | 226 | 108 | 2,767 |
| | Neutral | 1,671 | 129 | 71 | 1,871 |
| | Negative | 6,907 | 474 | 252 | 7,633 |
| | Mixed | 1,306 | 71 | 69 | 1,446 |
| Topic | Political | 4,037 | 279 | 159 | 4,475 |
| | Racial | 633 | 113 | 23 | 769 |
| | Sexist | 564 | 78 | 21 | 663 |
| | Religious | 244 | 157 | 11 | 412 |
| | Others | 6,839 | 273 | 286 | 7,398 |
| | **Total** | **12,317** | **900** | **500** | **13,717** |

GPT-4o [13] to ensure quality and consistency, creating a unique multimodal dimension for studying contextual factors in online abuse detection. See Appendix D for the model instructions and other details used in this step.

## 2.4   Inter-Annotator Agreement (IAA)

To compute IAA scores, we randomly sampled 100 comments from each annotator's contributions (a total of 900 samples) and then asked each of the nine annotators to provide secondary labels to comments that were not initially annotated by them. Then each comment in the sample was rated by two different annotators, resulting in a total of 1,800 annotations for each of the three tasks. This approach enabled us to compute the agreement between the pairs of contributors using Cohen's Kappa [14]. The aggregate scores for each task are: $\kappa = 0.758$ for *Abusiveness*, $\kappa = 0.649$ for *Sentiment*, and $\kappa = 0.603$ for *Topic* annotations. According to Cohen's interpretation, these scores indicate a *substantial agreement* for abusiveness detection and sentiment analysis annotations (i.e., $0.61 \leq \kappa \leq 0.80$) and a *moderate agreement* for topic classification (i.e., $0.41 \leq \kappa \leq 0.60$). We assessed 25 random comments that were assigned different Topic labels by the annotators and found that most disagreements were due to the potential applicability of the comments to multiple topics.

## 2.5   Gold-label Adjudication for Evaluation

To construct a high-quality test set, we extended the double-annotation process to three annotators and determined a gold label for each of the 900 samples. Our analysis showed that the initial two annotators achieved a full agreement on 546 comments, while they disagreed on at least one of the three tasks for the remaining 354 samples. To adjudicate the differences, we hired two additional experts who reviewed the inconsistent annotations of the initial annotators and decided on the final labels, which are considered as the *gold* labels in our test set.

Finally, we partitioned the remaining single-annotated samples into two splits for training and validation (500 samples), maintaining stratified proportions of the classes for abusiveness, sentiment, and topic in each split. Table 1 presents the detailed sample distribution across the splits, and Figure 1b depicts the overall class distribution of the TiALD dataset for the three tasks.

## 3   Experimental Setup of Baselines

To establish strong baselines for abusive language detection in Tigrinya, we evaluate three complementary approaches. First, we conduct single-task fine-tuning by training and evaluating several pre-trained language models (PLMs) on each classification task individually. Second, we implement a multi-task joint learning framework, where a shared encoder model simultaneously learns all three tasks through task-specific output heads. Finally, we assess the zero- and few-shot capabilities of state-of-the-art generative LLMs on the abusiveness detection task using prompts.

## 3.1 Single-task Fine-tuning

For each task in TiALD, we fine-tune several monolingual and multilingual PLMs that offer varying levels of adaptation to Tigrinya and other African languages. These include: monolingual models **TiRoBERTa** (125M) and **TiELECTRA** (14M) [15] trained exclusively on Tigrinya texts; multilingual models **AfriBERTa-base** (112M) [16] and **AfroXLMR-Large-76L** (560M) [17], pre-trained on 11 and 76 African languages, respectively; and **XLM-RoBERTa-base** (279M) [18], a general-purpose multilingual model pre-trained on 100 languages, as a control.

## 3.2 Joint Multi-task Training

In our joint learning setup, we employ a single transformer encoder shared across the three tasks that simultaneously learns to categorize the input content according to all relevant labels. Formally, let $\mathbf{h} = \text{Encoder}(x) \in \mathbb{R}^d$ denote the contextualized representation of an input comment $x$, given by the final hidden state corresponding to the model's classification token (e.g., '[CLS]' for BERT or '' for RoBERTa). A single linear classification head then maps $\mathbf{h}$ to a vector of logits:

$$\mathbf{z} = W\,\mathbf{h} + \mathbf{b}, \quad W \in \mathbb{R}^{L \times d}, \ \mathbf{b} \in \mathbb{R}^L,$$

where $L = 2 + 4 + 5 = 11$ is the total number of labels covering (i) abusiveness (binary), (ii) sentiment (4-way), and (iii) topic (5-way) tasks in the TiALD dataset. Each logit $z_j$ is passed through a sigmoid to produce the probability of label $j$; a threshold of $0.5$ is applied during inference to obtain binary predictions:

$$\hat{y}_j = \sigma(z_j), \quad j = 1, \ldots, L.$$

Training minimizes the average binary cross-entropy (BCE) loss over all label-example pairs:

$$\mathcal{L} = -\frac{1}{N} \sum_{i=1}^{N} \sum_{j=1}^{L} \Big[ y_{ij} \log \hat{y}_{ij} \ + \ (1 - y_{ij}) \log\big(1 - \hat{y}_{ij}\big) \Big],$$

where $N$ is the total number of examples in the training set and $y_{ij} \in \{0, 1\}$ indicates the presence of label $j$ in the $i$th example.

This hard-parameter-sharing approach treats each label as an independent binary predictor while leveraging a shared representation across all three tasks, encouraging the model to capture features that benefit abusive language detection, topic classification, and sentiment analysis simultaneously. In our baseline setup, all labels contribute equally to the loss; future work may explore per-task or per-class weighting to address label imbalance.

**Model training settings.** For single-task and multi-task fine-tuning experiments, we set the maximum input length to 256 tokens when using comment text only and 384 tokens when incorporating video titles. We use a learning rate of $2e^{-5}$, a batch size of 16, and train for a maximum of six epochs with early stopping based on validation macro F1 score (patience of 3). We employ the AdamW optimizer [19] and implement our training system with PyTorch [20] and the Hugging Face Transformers library [21].

## 3.3 In-context Learning of LLMs

To assess the capabilities of state-of-the-art generative LLMs on Tigrinya abusive language detection, we employ prompt-based zero- and few-shot in-context learning [22, 23]. We design prompt templates that include either no examples (zero-shot) or a small set of annotated examples (few-shot) randomly sampled from the training set. We evaluate two commercial frontier models, **GPT-4o** [13] and **Claude Sonnet 3.7** [24],[3] and two smaller open-weight models, **LLaMA-3.2 3B** [25] and **Gemma-3 4B** [26], for comparison. To account for variability, we run the predictions twice and compute the average scores. All input comments are preserved in their original script (Ge'ez, Latin, or mixed), and the prompt explicitly mentions that the comment is in Tigrinya. See Appendix C for the full template of the instructions used in our experiments.

---

[3] OpenAI's GPT-4o (gpt-4o-2024-08-06) and Anthropic's Sonnet 3.7 (claude-3-7-sonnet-20250219).

Table 2: Performance of fine-tuned encoder models (single and multi-task) and prompted generative LLMs (zero-shot and few-shot) evaluated on user comments across all three tasks. The *TiALD Score* is the average macro F1 across the three tasks. Overall task-level best scores are in **bold**; category-best scores are underlined.

| Model | Abusiveness | Sentiment | Topic | TiALD Score |
|---|---|---|---|---|
| **Fine-tuned Single-task Models** | | | | |
| TiELECTRA-small | 82.33 | 42.39 | 26.90 | 50.54 |
| TiRoBERTa-base | **86.67** | 52.82 | 54.23 | 64.57 |
| AfriBERTa-base | 83.42 | 50.81 | 53.20 | 62.48 |
| Afro-XLMR-Large-76L | 85.20 | **54.94** | 51.42 | 63.86 |
| XLM-RoBERTa-base | 81.08 | 30.17 | 43.97 | 51.74 |
| **Fine-tuned Multi-task Models** | | | | |
| TiELECTRA-small | 84.21 | 43.44 | 29.27 | 52.30 |
| TiRoBERTa-base | 86.11 | 53.41 | **54.91** | **64.81** |
| AfriBERTa-base | 83.66 | 50.19 | 53.49 | 62.45 |
| Afro-XLMR-Large-76L | 85.44 | 54.50 | 52.46 | 64.13 |
| XLM-RoBERTa-base | 79.87 | 45.40 | 35.50 | 53.59 |
| **Zero-shot Prompted LLMs** | | | | |
| GPT-4o | 71.05 | 20.55 | 26.25 | 39.28 |
| Claude Sonnet 3.7 | 59.20 | 22.64 | 25.25 | 35.70 |
| Gemma-3 4B | 59.35 | 29.47 | 35.24 | 41.35 |
| LLaMA-3.2 3B | 49.98 | 25.30 | 16.55 | 30.61 |
| **Few-shot Prompted LLMs** | | | | |
| GPT-4o | 72.06 | 21.88 | 27.56 | 40.50 |
| Claude Sonnet 3.7 | 79.31 | 23.39 | 27.92 | 43.54 |
| Gemma-3 4B | 58.37 | 30.46 | 39.49 | 42.78 |
| LLaMA-3.2 3B | 45.65 | 19.94 | 21.68 | 29.09 |

**Evaluation Metrics.** We report Macro F1 as the primary task-level metric. We prioritize F1 over accuracy due to the inherent class imbalance and the multi-class nature of the sentiment (4-way) and topic (5-way) classification tasks. To facilitate holistic comparison at the benchmark level, we introduce the *TiALD Score*, defined as the average of the task-level macro F1 scores. We supplement these metrics with per-class F1 scores to enable granular analysis of model performance, particularly on minority classes.

Our experimental setup provides strong baselines for comparing single and multi-task fine-tuning against in-context learning of LLMs for abusive language detection under low-resource settings.

## 4 Results and Analysis

### 4.1 Performance of Fine-tuned Encoder Models

Our experimental results demonstrate that jointly training models on all three tasks consistently enhances performance over the single-task approaches. This improvement suggests that abusiveness, sentiment, and topic share complementary linguistic features that benefit from unified representation learning. As shown in Table 2, Tigrinya-specific models outperform general multilingual alternatives. TiRoBERTa-base achieves the highest macro F1 scores across most settings (86.67% for abusiveness detection, 54.23% for topic classification), demonstrating the value of language-specific pre-training. The Africa-centric AfroXLMR-76L model performs competitively, particularly for sentiment analysis, where it reaches the highest F1 score of 54.94%, suggesting that well-adapted multilingual models can reach monolingual performance.

Multi-task joint learning improves performance across almost all models and tasks, with the most substantial gains observed for TiELECTRA-small (+1.76 percentage points in overall TiALD score) and XLM-RoBERTa-base (+1.85 points). The consistent improvement across diverse model architectures confirms that the complementary signals in the TiALD tasks can be leveraged through parameter sharing. Notably, XLM-RoBERTa-base consistently underperformed compared to both the Tigrinya-specific and Africa-adapted models, with a substantial 12 percentage point average gap

Table 3: Performance of models with video title as context. Fine-tuned models were trained on concatenation of user comment and video title. LLMs were prompted with both comment and video title. Overall task-level best scores are in **bold**; category-best scores are underlined.

| Model | Abusiveness | Sentiment | Topic | TiALD Score |
|---|---|---|---|---|
| **Fine-tuned Single-task Models** | | | | |
| TiELECTRA-small | 81.67 | 39.40 | 27.81 | 49.62 |
| TiRoBERTa-base | **86.17** | **54.97** | **54.55** | **65.23** |
| AfriBERTa-base | 82.44 | 51.33 | 52.10 | 61.96 |
| Afro-XLMR-Large-76L | 84.20 | 52.64 | 54.11 | 63.65 |
| XLM-RoBERTa-base | 75.09 | 43.47 | 41.60 | 53.39 |
| **Zero-shot Prompted LLMs** | | | | |
| GPT-4o | 75.59 | 41.03 | 55.52 | 57.38 |
| Claude Sonnet 3.7 | 67.64 | 44.39 | 50.10 | 54.05 |
| Gemma-3 4B | 58.41 | 29.27 | 34.44 | 40.71 |
| LLaMA-3.2 3B | 44.13 | 21.85 | 15.91 | 27.30 |
| **Few-shot Prompted LLMs** | | | | |
| GPT-4o | 75.89 | 45.50 | 58.59 | 59.99 |
| Claude Sonnet 3.7 | 80.29 | 48.01 | 59.45 | 62.58 |
| Gemma-3 4B | 59.39 | 30.43 | 39.60 | 43.14 |
| LLaMA-3.2 3B | 48.29 | 20.19 | 20.20 | 29.56 |

on the aggregate macro F1 scores. This performance disparity highlights the limitations of general multilingual models when applied to low-resource languages with unique linguistic characteristics.

## 4.2 Performance of Large Language Models

The results in Table 2 reveal that even state-of-the-art LLMs struggle to match the performance of small fine-tuned models on Tigrinya abusive language detection. GPT-4o achieves 71.05% F1 with zero-shot prompting, which, while impressive, still falls 15 percentage points behind the tuned TiRoBERTa-base. The performance gap between zero-shot and few-shot settings varies dramatically across models. Claude Sonnet 3.7 shows substantial improvement (59.20% → 79.31%), demonstrating high sensitivity to in-context examples. By contrast, the smaller open-weight models exhibit severe limitations in understanding Tigrinya text. LLaMA-3.2 3B shows classification bias that reverses across prompting conditions: in zero-shot settings, it classified 68% of comments as *abusive*, while in few-shot settings it conversely assigned 77% of them to *not abusive*, resulting in F1 scores of 49.98% and 45.65%, respectively. This inconsistency highlights the fundamental challenges current LLMs face when processing low-resource languages outside their primary training distribution.

**Task-Specific Performance Gaps.** Further analysis of the scores in Table 2 reveals a critical finding: while fine-tuned models maintain strong performance across all tasks (52-87% F1), LLMs exhibit severe degradation on multi-class sentiment and topic classification. The best LLM achieves only 30.46% F1 on sentiment and 39.49% on topic, showing significant deficits of 24 and 15 percentage points respectively compared to fine-tuned models. This disparity persists despite competitive LLM performance on binary abusiveness detection (71-79% F1), suggesting that current LLMs fundamentally struggle with fine-grained multi-class classification in low-resource settings. Interestingly, the smaller Gemma-3 4B outperforms frontier models on sentiment and topic tasks, indicating that model scale alone cannot overcome these limitations.

## 4.3 Impact of Contextual Information on Performance

Social media comments often respond to the original post, making contextual information valuable for understanding them. When video titles are added as context, the performance improves for most models, as shown in Table 3. TiRoBERTa-base trained on comments and video titles achieves the highest overall performance (65.23% TiALD score), with a significant gain (+1.5 points) observed in sentiment classification. This suggests that the video context provides topical cues that help disambiguate the intent and subject of the comments. While the generated video descriptions provided

Table 4: Performance of LLMs on Abusiveness Detection with Cross-Modality Contextual Information: user `comment` augmented with `video_title` and auto-generated `video_description`. Best scores for each prompting approach are in **bold**; highest scores within model category are underlined.

| | Comment Only | | Video Title + Comment | | Video Title + Desc. + Comment | |
|---|---|---|---|---|---|---|
| | Zero-shot | Few-shot | Zero-shot | Few-shot | Zero-shot | Few-shot |
| **Closed Frontier Models** | | | | | | |
| GPT-4o | **71.05** | 72.06 | **75.59** | 75.89 | **74.70** | 74.53 |
| Claude Sonnet 3.7 | 59.20 | **79.31** | 67.64 | **80.29** | 72.02 | **78.21** |
| **Open-weight Models** | | | | | | |
| Gemma-3 4B | 59.35 | 58.37 | 58.41 | 59.39 | 54.84 | 50.95 |
| LLaMA-3.2 3B | 49.98 | 45.65 | 44.13 | 48.29 | 48.64 | 29.44 |

valuable contextual signals for LLMs with their large context windows (Table 4), incorporating this long-form text into fine-tuned encoder models was not feasible due to their limited input length constraints of 256-512 tokens.

More detailed class-level breakdown of model performances can be found in Appendix. Tables 5 and 6 present per-class F1 scores across all experimental settings.

## 4.4 Analysis and Insights

**Cross-Task Performance Analysis.** Performance analysis reveals that models achieve higher F1 scores for detecting abusive content (79-86%) compared to sentiment analysis (30-54%) and topic classification (26-54%). This pattern likely reflects the multi-class nature of sentiment and topic annotations, increasing the difficulty of the tasks, as evidenced by the lower inter-annotator agreement scores. The best-performing model, TiRoBERTa-base with the joint learning setup, demonstrates a relatively balanced increase across the sentiment (+0.6) and topic (+0.7) tasks, but both remain challenging with only 53.41% and 54.91% macro F1 scores, respectively. This performance gap presents an opportunity for future research to develop more specialized approaches to sentiment and topic understanding in morphologically complex languages like Tigrinya.

**Effectiveness of Iterative Seed-Expansion Sampling.** Compared to conventional fixed-vocabulary methods, our iterative seed-expansion sampling approach generates a more diverse and representative annotation pool while preserving lexical diversity, which is a crucial factor for languages with highly inflectional morphology where words can take numerous surface forms. Quantitative analysis reveals that our approach yielded a higher type-to-token ratio (27.6%) compared to keyword-based sampling using existing toxic word lists (18.2%) and the source corpus (7.2%) of all the 4.1M comments. Furthermore, we analyzed the ratio of *abusive* class annotations for the comments from our iterative sampling against those in the control group via random sampling, and we observed a significant difference, 65.2% vs. 14.3%, respectively. The random sampling is biased towards the majority type and hence leads to more benign non-abusive comments, while the keyword sampling is biased towards the seed words and fails to produce a diverse pool of samples.

**Cross-Modality Context for Abusiveness Detection in LLMs.** As shown in Table 4, the performance of the frontier LLMs improves when the user comments are enriched with contextual information (i.e., video titles and the auto-generated video descriptions). GPT-4o performs the best in the zero-shot settings, gaining 3.65 percentage points. Similarly, Claude Sonnet 3.7 shows substantial improvement in zero-shot performance (+12.82 points) when provided with video context. These gains underscore the importance of contextual understanding in accurately identifying abusive content, particularly when the language itself presents challenges for the models. The consistent improvement across settings confirms our hypothesis that supplementary video context provides valuable signals for content moderation in low-resource languages.

## 5 Related Work

**Datasets for Abusive Language Detection.** Numerous datasets have been created for the purpose of training and evaluating models for abusive language detection in English, typically sourced from

online platforms such as Twitter, Reddit, YouTube, and Wikipedia [27–31]. Researchers have also developed datasets for languages other than English, such as the Dutch-Bully-Corpus [32] consisting of over 85,000 abusive posts, and the Arabic dataset by Mubarak et al. [33] containing 1,100 tweets and 32,000 YouTube comments. Moreover, shared tasks such as OffensEval [5, 34], GermEval 2018 [35], and HASOC 2019 [36] have provided multilingual datasets for such tasks. Pavlopoulos et al. [37] created the Greek-Gazzetta-Corpus, consisting of 1.6 million comments, and Song et al. [38] proposed a comprehensive abusiveness detection dataset with multifaceted labels from Reddit. Furthermore, Tonneau et al. [39] introduced a dataset for hate speech detection containing 35,976 tweets, comprising instances of mixed languages such as English, Pidgin, Hausa, and Yoruba.

**Approaches to Abusive Language Detection.** Approaches to abusive language detection have evolved from statistical models to deep neural architectures, with transformer-based models like BERT, RoBERTa, and the GPT-family, establishing the state-of-the-art performance across multiple benchmarks [40–45]. For low-resource languages, cross-lingual transfer from multilingual models offers significant benefits [46, 34]. Recent advances applied generative Large Language Models (LLMs) with in-context learning and data augmentation of abusive language [22, 47]. For instance, Shin et al. [48] demonstrated that GPT-generated synthetic data enhanced offensive language detection in Korean. Zhang et al. [49] proposed an approach of Bootstrapping and Distilling LLMs for toxic content detection using a novel Decision-Tree-of-Thought prompting. Jaremko et al. [50] evaluated the performance of LLMs for implicitly abusive language detection using zero-shot and few-shot learning approaches, and analyzed the models' ability to extract relevant linguistic features.

**Multi-Task Learning for Abusiveness Detection.** Recent studies have demonstrated the efficacy of multi-task learning (MTL) in enhancing abusive language detection by jointly modeling related tasks. For instance, Dai et al. [51] employed a BERT-based MTL framework to simultaneously address offensive language detection, categorization, and target identification, achieving promising results. Similarly, Zhu et al. [52] integrated Prompt tuning [22, 53] with MTL to improve detection performance across multiple datasets. Mnassri et al. [54] leveraged MTL to jointly model hate speech and emotions, showing that sentiment information improves hate speech detection performance. In a related approach, Rajamanickam et al. [55] demonstrated that joint learning of toxicity and sentiment leads to more robust models than single-task approaches.

**Tigrinya Language Processing.** Tigrinya (ISO 639-3: `tir`) is a Semitic language of the Afro-Asiatic family that shares linguistic features with Amharic and Tigre, and uses the Ge'ez script as a writing system [9]. There is a growing interest in computational approaches to Tigrinya, such as machine translation, part-of-speech tagging, sentiment analysis, text classification through transfer learning [56–61, 9]. More recent progress on question answering and named entity recognition [62, 63] has been enabled by datasets and pre-trained language models [64, 15, 16].

**Resources for African Languages.** In a related effort, Ayele et al. [65] developed a dataset for Amharic, which contains 8,258 tweets annotated for hate/offensive category, target type, and intensity on a continuous scale. Contemporary to our work, Muhammad et al. [7] introduced AfriHate, a large-scale Twitter-based benchmark dataset for hate and abusive language in 15 African languages, including a Tigrinya (tir) subset with 5,072 tweets annotated for abusiveness and target types. The Tigrinya slice shows a class imbalance typical of keyword-retrieval pipelines and a moderate inter-annotator agreement. Baseline experiments on AfriHate demonstrate that multilingual fine-tuned models reach a macro F1 of 74.5%, outperforming few-shot prompting of GPT-4o [13] by 18 points. In TiALD, we employ a semi-automated data-driven sampling strategy designed to diversify the annotation pool, resulting in a more challenging and representative benchmark. We sourced data from YouTube due its substantially higher popularity among Tigrinya speakers in Eritrea and Ethiopia compared to Twitter (X) and other social media platforms.[4] Furthermore, YouTube comments are accompanied by rich context, such as video title and description, beneficial for modeling abusive content detection. Thus, TiALD and AfriHate are complementary resources for the research community with different characteristics. To the best of our knowledge, there is no prior resource that simultaneously addresses abusiveness, sentiment, and topic under low-resource settings.

---

[4] YouTube vs. Twitter (X) usage in Eritrea (36.4% vs. 9.1%) and Ethiopia (13.1% vs. 6.8%) since 2018: https://gs.statcounter.com/social-media-stats/all/eritrea/#monthly-201801-202503 and https://gs.statcounter.com/social-media-stats/all/ethiopia/#monthly-201801-202503.

# 6  Conclusion

In this work, we introduced the first large-scale multi-task benchmark dataset for abusive language detection in Tigrinya. **Ti**grinya **A**busive **L**anguage **D**etection (TiALD) dataset comprises 13,717 user comments from YouTube videos, annotated by native speakers for three tasks: abusiveness, sentiment, and topic classification, providing a rich resource for understanding and detecting harmful content in this understudied language's social media. Our comprehensive experiments establish strong baselines while revealing substantial room for improvement across all tasks, with performance gaps of 15-45 percentage points from perfect classification, indicating that TiALD will serve as a challenging benchmark for future research. Our analysis reveals three key insights: first, we demonstrate that small, specialized Tigrinya models substantially outperform the current frontier models in the low-resource setting; second, we show that joint multi-task learning in aggregate outperforms single-task approaches, indicating that the abusiveness, sentiment, and topic tasks share complementary signals; third, incorporating auxiliary visual content descriptions further enhances abusiveness detection performance. We make the TiALD dataset and trained models publicly available to advance content moderation research for the Tigrinya-speaking community and to serve as a blueprint for similar efforts in other low-resource languages, promoting more inclusive and effective online safety systems.

## Limitations

**Explicit *vs.* Implicit Abusiveness:** As the first study of its kind for Tigrinya, our work focuses on overt forms of offensive language. We acknowledge that implicit forms of toxicity, such as microaggressions and subtle forms of prejudice, also contribute to online harassment and should be addressed in future work. **Granular Annotation of Abusiveness:** Our dataset includes a single label for abusiveness, which may not capture the full range of abusive language. Future work could look into a more granular annotation scheme that captures nuanced subtypes of abusive language.

## Ethics Statement

This research adheres to the academic and professional ethics guidelines of our institution, obtaining its Institutional Review Board (IRB) approval.[5] All data collection and annotations were conducted with informed consent of the participants. While the development of abusive content detection systems has the potential to improve the online experience of millions of social media users worldwide, it is crucial to consider the possible societal and ethical implications of such research. **Fairness and bias mitigation:** We carefully designed data collection and annotation procedures to minimize biases and avoid reinforcing stereotypes in the dataset and baseline models. **Respecting privacy:** We adhere to strict privacy guidelines while collecting and using user-generated data, ensuring that any personal information remains anonymized and protected. **Balancing moderation and expression:** We recognize the tension between detecting harmful content and protecting free expression, emphasizing that systems built on our dataset should incorporate transparent, accountable processes to minimize over-censorship. By addressing these considerations and making our dataset publicly available, we aim to contribute to safer online environments for Tigrinya speakers while providing a foundation for ethical content moderation research.

## Acknowledgments and Disclosure of Funding

We would like to appreciate the language communities that contributed to this work through annotation, validation, and feedback. We also thank the anonymous reviewers for their valuable input. This work was supported by the National Research Foundation of Korea (NRF) grant funded by the Korea government (MSIT) (No. RS-2023-00208054). This work was also supported by the GeezLab Research Program funded by GeezLab.com (No. 2023-001).

---

[5] Approval number: KH2022-133

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

# Appendix

## A    Class-level Performance of Baseline Models on the TiALD Benchmark

We analyze model classification biases through the lens of performance variance across the three tasks in TiALD and data subsets, revealing critical insights for content moderation in low-resource settings. The class-level results expose disparities that have important implications for real-world deployment.

### A.1    Analysis of Fine-tuned Encoder Models

Table 5 presents the per-class F1 scores for all three tasks. For abusiveness detection, performance is relatively balanced between *abusive* and *not abusive* classes, with TiRoBERTa-base achieving the highest scores (86.52% and 86.81% in the single-task setting). However, sentiment and topic classification reveal severe class imbalances that reflect both natural data distribution and the inherent difficulty of nuanced content classification.

For sentiment analysis, all models exhibit strong bias toward the *negative* class (achieving up to 81.32% F1) while dramatically underperforming on *neutral* and *mixed* classes (as low as 1.53% and 0% F1). This disparity suggests that models default to negative sentiment when uncertain, potentially over-flagging neutral content in production systems.

Topic classification shows similar patterns, with models achieving strong performance on *political* content (up to 71.21% F1) but substantially weaker results on minority classes. Most concerning are the near-zero F1 scores for *racial* (6.50%), *sexist* (0%), and *religious* (0%) categories in some single-task configurations, indicating complete failure to identify these sensitive content types.

Critically, multi-task joint learning demonstrates significant bias mitigation. TiRoBERTa-base's F1 score on *sexist* content improves dramatically from 31.78% to 46.30% (+14.52 points) with joint learning. Similarly, performance on *neutral* sentiment increases from 40.0% to 42.75%, and *religious* content shows recovery from complete failure in TiELECTRA (0% to 7.36%). These improvements demonstrate that the complementary signals across tasks help models better recognize minority classes, a crucial benefit for equitable content moderation.

Table 5: Class-level Performance of Single and Multi-task settings in F1 score. Models are trained and evaluated on the comment text only. The highest class-level scores for each approach are underlined, and the overall best scores are in **bold**. Multi-task learning yields significant performance improvements for the minority classes (e.g., *sexist*, *religious*).

| Model | Abusiveness | | Sentiment | | | | Topic | | | | |
|---|---|---|---|---|---|---|---|---|---|---|---|
| | Abusive | Not Abusive | Positive | Neutral | Negative | Mixed | Political | Racial | Sexist | Religious | Other |
| **Single-task Models** | | | | | | | | | | | |
| TiELECTRA-small | 82.35 | 82.31 | 62.84 | 4.48 | 80.88 | 21.36 | 67.48 | 06.50 | 00.00 | 00.00 | 60.51 |
| TiRoBERTa-base | **86.52** | **86.81** | 68.68 | 40.00 | 81.18 | 21.43 | 70.86 | 46.49 | 31.78 | 56.90 | 65.15 |
| AfriBERTa-base | 84.00 | 82.85 | 63.25 | 33.78 | 81.03 | 25.17 | 69.64 | 40.46 | 34.29 | 56.89 | 64.71 |
| Afro-XLMR-Large-76L | 85.74 | 84.66 | 69.44 | 41.06 | **81.32** | 27.94 | 70.71 | 39.75 | 28.00 | 54.38 | 64.29 |
| XLM-RoBERTa-base | 80.32 | 81.84 | 46.03 | 01.53 | 73.11 | 00.00 | 67.32 | 33.33 | 02.53 | 52.53 | 64.12 |
| **Multi-task Joint Learning** | | | | | | | | | | | |
| TiELECTRA-small | 84.67 | 83.75 | 62.68 | 14.39 | 81.10 | 15.58 | 65.17 | 06.84 | 07.41 | 07.36 | 59.56 |
| TiRoBERTa-base | 86.13 | 86.10 | 62.98 | **42.75** | 79.83 | 28.07 | **71.21** | 45.96 | **46.30** | 47.44 | 63.65 |
| AfriBERTa-base | 83.93 | 83.39 | 65.39 | 27.43 | 81.13 | 26.79 | 70.59 | 44.16 | 44.00 | 43.35 | **65.36** |
| Afro-XLMR-Large-76L | 85.16 | 85.71 | 71.79 | 33.88 | 80.96 | **31.34** | 69.02 | 36.60 | 35.64 | 57.78 | 63.27 |
| XLM-RoBERTa-base | 80.43 | 79.31 | 67.06 | 15.47 | 79.92 | 19.15 | 67.51 | 16.26 | 16.47 | 15.29 | 61.95 |

### A.2    Performance Disparity in Large Language Models

Class-level evaluation on the TiALD tasks reveals striking performance disparities in the generative LLMs, as shown in Table 6. Despite the balanced test set (50% samples per class of abusiveness), models show severe classification imbalances. Claude Sonnet 3.7 exhibits a highly asymmetric zero-shot performance (44.18% F1 on *abusive* vs. 72.58% on *not abusive*), suggesting a bias toward classifying content as non-abusive. However, it shows dramatic improvements in the few-shot setting, achieving 79.26% and 80.69% F1 for *abusive* and *not abusive* classes, respectively, but still falls short compared to fine-tuned small models. Adding contextual information partially mitigates the

disparities but also yields mixed results. GPT-4o's F1 score for detecting *abusive* content jumps from 69.04% to 76.66% when provided with video context.

LLaMA-3.2 3B exhibits dramatic classification instability across prompting conditions, predicting 68% of comments as *abusive* in zero-shot and inversely 77% as *not abusive* in few-shot settings. This erratic behavior stems from the model's fundamental limitation to comprehend Tigrinya text. Tokenization analysis reveals LLaMA-3.2 requires 2.31 tokens per character for Tigrinya versus only 0.20 for English (an 11.5× increase), significantly impacting both accuracy and inference cost for low-resource language deployment.

Table 6: Class-level Performance of LLMs across all tasks in TiALD evaluated on the user comment. The highest class-level scores for each approach are underlined, and the overall best scores are in **bold**. Most models show severe classification biases on the balanced test set. Reported in F1 score.

| Model | Abusiveness | | Sentiment | | | | Topic | | | | |
|---|---|---|---|---|---|---|---|---|---|---|---|
| | Abusive | Not Abusive | Positive | Neutral | Negative | Mixed | Political | Racial | Sexist | Religious | Other |
| **Zero-shot Prompted LLMs** | | | | | | | | | | | |
| GPT-4o | 69.04 | 75.89 | 51.03 | 14.55 | 76.16 | **22.68** | 62.96 | 33.70 | 27.78 | 75.07 | 63.04 |
| Claude Sonnet 3.7 | 44.18 | 72.58 | **65.85** | 29.80 | 77.94 | 07.55 | **66.90** | 21.33 | 19.78 | **79.00** | **65.48** |
| Gemma-3 4B | 59.25 | 60.94 | 35.95 | 00.00 | 69.49 | 12.00 | 55.03 | 03.45 | 06.19 | 64.31 | 49.62 |
| LLaMA-3.2 3B | 59.81 | 42.43 | 09.60 | 13.20 | 64.01 | 13.00 | 00.70 | 10.20 | 00.00 | 22.22 | 48.47 |
| **Few-shot Prompted LLMs** | | | | | | | | | | | |
| GPT-4o | 74.82 | 71.41 | 54.40 | 23.92 | 74.32 | 22.38 | 60.85 | 37.12 | 37.87 | 78.72 | 61.02 |
| Claude Sonnet 3.7 | **79.26** | **80.69** | 65.75 | **33.18** | **79.65** | 8.55 | 63.59 | **43.26** | **39.05** | 78.23 | 55.07 |
| Gemma-3 4B | 52.52 | 64.26 | 25.81 | 22.22 | 57.56 | 17.72 | 55.97 | 09.16 | 15.84 | 60.90 | 56.70 |
| LLaMA-3.2 3B | 28.30 | 60.16 | 26.67 | 20.29 | 15.52 | 16.13 | 21.14 | 06.76 | 11.43 | 23.40 | 48.76 |

## A.3 Script-Based Robustness and Joint Annotation Benefits

Our dataset's accommodation of both Ge'ez script and Romanized text (reflecting the 64% Romanized usage in real Tigrinya social media) enables models to develop script-agnostic representations. Initial qualitative analysis indicates that models trained on this mixed-script data show more robust performance across both writing systems, though comprehensive quantitative evaluation is reserved for future work.

Furthermore, the joint annotations in TiALD enable nuanced analysis beyond binary classification. Comments labeled as both *Abusive* and *Political* can be interpreted as political hate speech, while *Abusive*+*Sexist* combinations identify misogynistic content. This multi-dimensional labeling provides pathways for fine-grained content moderation without requiring extensive re-annotation, addressing reviewer concerns about granularity while maintaining high annotation quality.

## A.4 Implications for Low-Resource Content Moderation

The demonstrated performance disparities have critical implications for deploying content moderation systems in low-resource settings. The severe underperformance on minority classes means that certain types of harmful content (particularly sexist and religious abuse) may go undetected. Our results demonstrate that:

1. Multi-task learning provides a practical approach to mitigate these biases without additional data collection

2. Current LLMs, despite their impressive capabilities in high-resource languages, require fundamental architectural changes (particularly in tokenization) to serve low-resource languages effectively

3. Fine-tuned specialized models significantly outperform general-purpose LLMs, achieving 86.67% F1 compared to 79.26% for the best LLM configuration

# B TiALD Dataset Features

Table 7 presents the descriptions of the fields in the TiALD dataset. Figure 2 depicts the task-wise class distribution across the tasks of Abusiveness, Sentiment, and Topic.

Table 7: An overview of the features included in the TiALD Dataset.

| Feature | Data Type | Description |
| --- | --- | --- |
| sample_id | String | Unique identifier for the sample in the dataset. |
| comment_id | String | Unique identifier for the comment. |
| comment_original | String | Original comment text as posted by user. |
| comment_cleaned | String | Pre-processed version of the comment text. |
| abusiveness | Categorical | Abuse label (*Abusive* or *Not Abusive*). |
| sentiment | Categorical | Sentiment label (*Positive*, *Neutral*, *Negative*, or *Mixed*). |
| topic | Categorical | Topic label (*Political*, *Racial*, *Sexist*, or *Religious*, *Other*). |
| annotator_id | String | Identifier of the annotator. |
| comment_script | Categorical | Script used in the comment (*Ge'ez*, *Latin*, or *Mixed*). |
| comment_publish_date | String | Year and month the comment was published. |
| video_id | String | Identifier of the video the comment was posted under. |
| video_title | String | Title of the associated video. |
| video_num_views | Numeric | Number of views the video received. |
| video_publish_year | Numeric | Year the video was published. |
| video_description | String | Auto-generated description of the video content. |
| channel_id | String | Identifier of the YouTube channel the video belongs to. |
| channel_name | String | Name of the YouTube channel the video belongs to. |

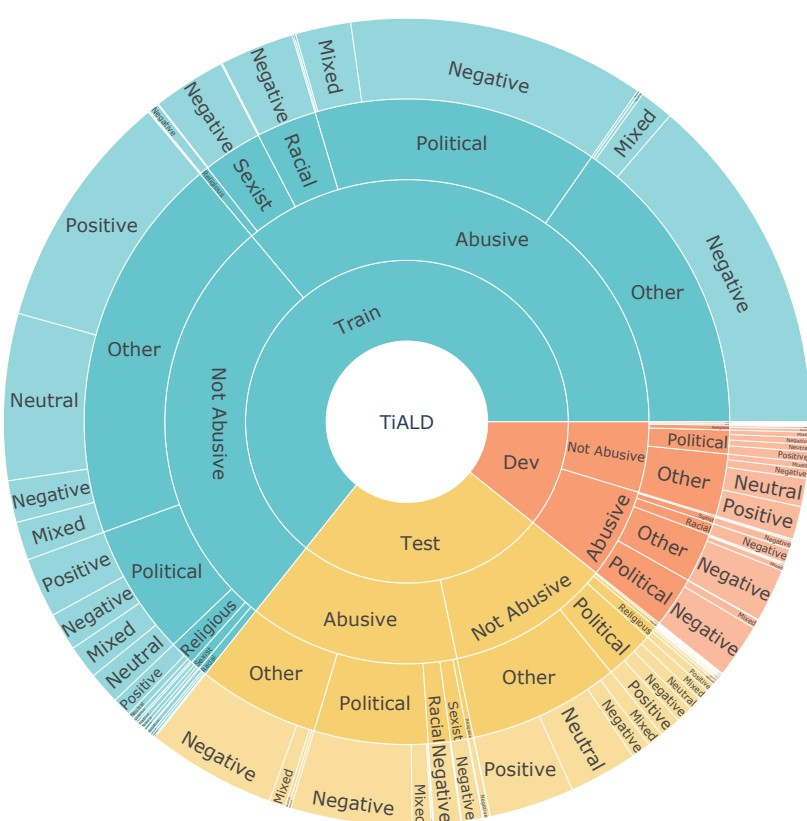

Figure 2: TiALD Class Distribution across the Splits and Tasks of Abusiveness, Sentiment, and Topic.

## C  Evaluation Instructions for LLMs

We conducted experiments on two commercial frontier models and two open-weight smaller models under both zero-shot and few-shot settings. All input comments were preserved in their original script (Ge'ez, Latin, or mixed), with an explicit instruction that the comment was written in Tigrinya. For the few-shot evaluation, we included four balanced examples (two abusive, two non-abusive) randomly sampled from the training set and arranged in alternating order as follows:

> [Instruction]
> ```
> Classify  the  following  user  comment  written  in  Tigrinya  as
> "Abusive"  or  "Not Abusive".   Do  not  provide  any  explanation  or
> additional text.
> ```
>
> [Optional In-context Examples]
> ```
> Here are some examples:
> <comment text>: Abusive
> <comment text>: Not Abusive
> <comment text>: Abusive
> <comment text>: Not Abusive
> ```
>
> [Comment]
> ```
> Comment: <comment text>
> ```

## D  Generating Video Content Descriptions

We used a vision-language model, Qwen2.5-VL-3B [12], to generate detailed descriptions of video content corresponding to comments in the evaluation splits of the TiALD dataset. Qwen2.5-VL handles long-form videos up to several hours by applying dynamic resolution processing and absolute time encoding. The TiALD dataset's videos average 28.1 minutes (1,686 seconds) in length and can run as long as 334.75 minutes (20,085 seconds) at 30 FPS. To ensure high-quality descriptions, we trimmed videos exceeding 20 minutes to their first 20-minute segment. Each resulting clip was then passed to the model with the following instruction:

> [Instruction]
> ```
> Describe  the  content  of  this  video  frame  in  detail.   Focus  on
> the  people,  objects,  actions,  and  settings  visible  in  the  image.
> Provide  a  comprehensive  description  that  could  help  understand
> what  the  video  is  about.
> ```
>
> [Video]
> ```
> <video frames>
> ```

To enhance the quality and consistency of the generated video descriptions, we revise them using a larger, more capable model, GPT-4o [13], with the following instruction:

> [Instruction]
> ```
> Revise  the  following  automatically  generated  video  description
> to  make  it  clearer  and  consistent,  while  preserving  the  key
> information.  Keep  your  response  to  a  moderate  length,  up  to  150
> words, and focus only on the video content.
> ```
>
> [Video Title]
> ```
> Video Title: <video title>
> ```
>
> [Video Description]
> ```
> Video Description: <video description>
> ```

Finally, the resulting video descriptions were included in the dataset as auxiliary features to enable deeper analysis of potential relationships between video content and the abusiveness of comments.

We also empirically show the benefit of using contextual information in our experiments, as shown by the results in Table 4.

# E   Annotation Guidelines for TiALD Dataset

This section outlines the detailed annotation guidelines provided to the native Tigrinya speakers who participated in the TiALD dataset creation. Annotators were instructed to classify each YouTube comment across three dimensions: Abusiveness, Sentiment, and Topic, while following specific protocols for comment eligibility.

**Task 1: Abusive Language Detection**

Annotators were asked to determine whether a comment contained abusive language according to the following criteria:

- **Abusive**: The comment contains language that attacks, insults, demeans, or threatens an individual or group. This includes hate speech, profanity directed at others, derogatory terms, threats of violence, severe personal attacks, or language intended to humiliate or degrade.

- **Not Abusive**: The comment does not contain language that attacks an individual or group. It may express disagreement, criticism, or negative opinions without using abusive language toward others.

**Task 2: Sentiment Analysis**

Annotators classified the emotional tone of each comment into one of four sentiment categories:

- **Positive**: The comment expresses primarily positive emotions, approval, praise, gratitude, happiness, or optimism. This includes congratulatory messages, expressions of joy, or positive feedback.

- **Neutral**: The comment does not express a clear positive or negative sentiment. This includes factual statements, questions without emotional content, or balanced objective comments.

- **Negative**: The comment expresses primarily negative emotions, disapproval, criticism, anger, sadness, or pessimism. This includes expressions of disappointment, frustration, or negative judgments.

- **Mixed**: The comment contains a relatively balanced mix of both positive and negative sentiments, with neither clearly dominating. This includes comments expressing contrasting emotions or evaluating different aspects both positively and negatively.

**Task 3: Topic Classification**

Annotators classified each comment into one of five topical categories:

- **Political**: Comments discussing political figures, parties, governments, policies, elections, or expressing political opinions. This includes references to specific political events, governance issues, or politically divisive topics.

- **Racial**: Comments referring to racial or ethnic identity, characteristics, or relationships between racial/ethnic groups. This includes discussions about cultural identity tied to ethnicity.

- **Sexist**: Comments discussing gender roles, gender identity, or containing gendered language. This includes content related to expectations based on gender or discussions about gender relations.

- **Religious**: Comments discussing religious beliefs, practices, institutions, or figures. This includes references to religious texts, doctrines, religious communities, or spirituality.

- **Other**: Comments that don't primarily fall into any of the above categories. This includes everyday conversations, entertainment, personal updates, or general information sharing.

**Comment Eligibility Criteria**

Annotators were instructed to exclude comments from annotation if they met any of the following conditions:

1. Comments written entirely in a language other than Tigrinya (e.g., English, Amharic, Arabic)
2. Comments containing no actual Tigrinya words (e.g., consisting only of repeated characters, symbols, or emojis)
3. Comments that were unintelligible or lacked meaningful content

However, annotators were instructed to retain comments if they:

- Contained at least some Tigrinya words, even if mixed with words from other languages
- Used Romanized Tigrinya (Latin script) rather than the native Ge'ez script
- Contained code-switching between Tigrinya and other languages

These guidelines were designed to create a dataset that accurately reflects the linguistic and cultural nuances of abusive language in Tigrinya social media content. Figure 3 shows the annotation system we developed to annotate the TiALD dataset.

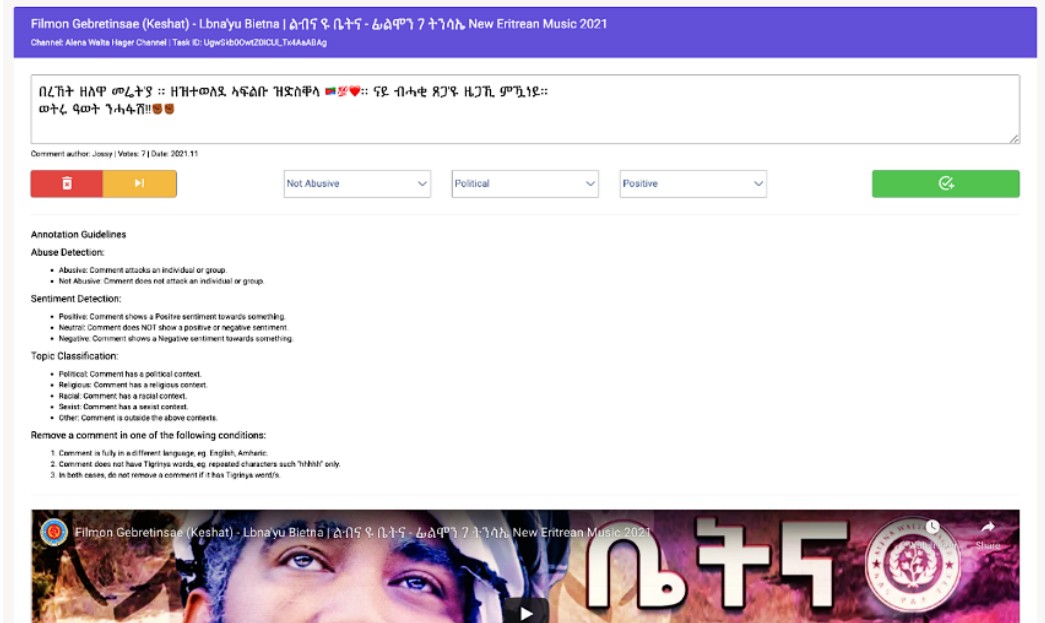

Figure 3: The annotation system we developed for the TiALD dataset. A summary of the annotation guidelines is provided on screen to encourage consistent annotations.

