# OpenReview forum: "A Multi-Task Benchmark for Abusive Language Detection in Low-Resource Settings"
_NeurIPS.cc/2025/Datasets_and_Benchmarks_Track — NeurIPS 2025 Datasets and Benchmarks Track poster_

### Official Review · Reviewer_36pD · 2025-06-17

**Rating:** 4
**Confidence:** 4

**Summary:**

This research introduces a large-scale, human-annotated multi-task benchmark dataset for detecting abusive language in Tigrinya social media. The dataset includes 13,717 YouTube comments annotated by nine native speakers for three tasks: abusiveness, sentiment, and topic classification. An iterative term clustering approach was developed for effective data selection. Recognizing that approximately 64% of Tigrinya social media content uses Romanized transliterations, the dataset accommodates both writing systems. Strong baselines were established, and the study found that small, specialized multi-task models outperform current state-of-the-art models in low-resource settings, achieving up to 86% accuracy in abusiveness detection. The resources are made publicly available to foster research on online safety.

**Dataset Code Accessibility:**

Yes

**Ethical Considerations:**

No, there are no or only very minor ethics concerns

**Final Justification:**

My doubts have been resolved.

**Limitations Weaknesses:**

- As the authors mentioned in Section 3.5, 60% of the samples have completely consistent criteria. It is recommended that the authors display the distribution of inconsistent criteria, as this is crucial for the results.

- The comparison of baseline models is insufficient.

- I noticed in Table 3 that for the Abusiveness detection task, Claude Sonnet 3.7 performs few-shot learning close to the model performance in Table 2. However:
  - In Table 3, why does the few-shot performance of Claude Sonnet 3.7 decrease after adding Video Context in the closed-source model, while the performance of other models increases?
  - In Table 3, why does the performance of open-source models decrease after adding Video Context?

- Although the authors set up a low-resource experimental scenario, as an academic study, I believe it is necessary to supplement the results with scenarios that have better computational resources, such as open-source models of 7B, 13B, and 32B.

- I noticed that the authors used LLMs only for the Abusiveness detection task. Additional experimental results are also needed for the other two tasks, as well as for multi-task detection experiments based on LLMs.

- For LLMs, the authors only conducted zero-shot and few-shot experiments. Could some open-source LLM models be fine-tuned to evaluate the experimental results? I suspect that the poor performance of large models is due to language understanding issues, and I hope the authors can verify this.

In summary, there are too few experimental results for large models, which are insufficient for a comprehensive evaluation of their capabilities in this task. It is recommended to supplement the experiments with large models to facilitate future research.

**Strengths Contributions:**

- The author focuses on low-resource multi-task detection for less common languages, covering aspects such as hate speech and sentiment analysis. This approach is innovative.
- The use of word vector methods ensures data diversity.
- IAA is employed to measure the consistency of data annotation.

---

> ### Author Rebuttal · Authors · 2025-07-28
>
> Thank you for your time and valuable suggestions, which will help us improve the clarity of our paper.
>
> > **Review:** "As the authors mentioned in Section 3.5, 60% of the samples have completely consistent criteria. It is recommended that the authors display the distribution of inconsistent criteria..."
>
> We would like to clarify our gold-label creation process. The figure of ~60% (546 out of 900 samples) refers to the initial agreement between the first two annotators on the test set *before adjudication*. The remaining 354 samples with disagreements were then resolved by expert annotators to create a single high-quality gold standard label (for each of the three tasks per sample) for 100% of the test set. We apologize for the lack of clarity and will revise Section 3.5 to make this process more explicit.
>
> > **Review:** "...the authors used LLMs only for the Abusiveness detection task. Additional experimental results are also needed for the other two tasks..."
>
> Thank you for pointing this out. For the generative LLMs, we focused our initial experiments on the primary task of abusiveness detection due to the high computational cost required for multi-task generation with the larger models and also the need for robust answer parsing required for each task. We will include the results for the sentiment and topic classification tasks for LLMs in the final version of the paper, as we did for the the encoder models in Tables 2 and 4.
>
> > **Review:** "Could some open-source LLM models be fine-tuned to evaluate the experimental results? I suspect that the poor performance of large models is due to language understanding issues, and I hope the authors can verify this."
>
> This is an excellent observation and suggestion. We agree that the performance of existing open-source LLMs is hindered by a fundamental lack of understanding of the target language. We were able to support this through our analysis of the tokenization efficiency. The prominent open-weight models are under-optimized for Tigrinya, as they may not have seen sufficient content during training of the tokenizer and pre-training of the model, leading to suboptimal processing.
>
> While we agree that fine-tuning open-source LLMs represents a promising future direction, along with potential *vocabulary expansion* and *continued pre-training*, we believe such an extensive undertaking is beyond the scope of the baseline systems this work aims to establish. Our primary contribution in this paper is the construction and introduction of a novel benchmark dataset. It is precisely this benchmark that we hope will enable and inspire future research into the very adaptation techniques, such as those suggested.

---

### Official Review · Reviewer_9xmK · 2025-06-29

**Rating:** 5
**Confidence:** 4

**Summary:**

This paper introduces TiALD, a multi-task benchmark dataset for abusive language detection in Tigrinya, featuring 13,717 YouTube comments annotated for abusiveness, sentiment, and topic classification. The authors address data scarcity by proposing an iterative term-clustering method for balanced sampling and accommodate Romanized script (64% of Tigrinya social media content). Experiments show specialized multi-task models outperform large LLMs, achieving 86% accuracy in abusiveness detection. The dataset and code are publicly released.

**Additional Feedback:**

1. The single linear head for all 11 labels (across 3 tasks) may limit task-specific learning. Why don't you use three task-specific heads?

**Dataset Code Accessibility:**

Yes

**Ethical Considerations:**

No, there are no or only very minor ethics concerns

**Final Justification:**

The author provided more experiments to discuss the  model biases across different data subsets.

**Limitations Weaknesses:**

* Unclear implementation details:
    - Section 3.2: It is not clear to how they handle code-mixing contents when they use word2vec. Which word2vec did they use? The iterative expansion process lacks algorithmic details (e.g., distance metrics for word embeddings, criteria for selecting k neighbors).
* Video titles improve performance (Table 4), but generated video descriptions are only used for LLMs, not fine-tuned models. Why don't Integrate video descriptions into fine-tuning?
* Training details for monolingual models (TiRoBERTa, TiELECTRA) are omitted. What are the pre-training data size, domains, and hyperparameters?
*  No discussion of model biases (e.g., performance variance across topics/scripts).

**Strengths Contributions:**

* The dataset addresses a critical gap in low-resource NLP by focusing on Tigrinya, spoken by 10M+ people but severely under-resourced.
* The dataset seems be high-quality and diverse. Comments from 7,373 videos (1.2B+ views) covering both Ge’ez and Latin scripts. It was annotated with 9 native speaker with substantial inter-annotator agreement (κ=0.758 for abusiveness). They also used Qwen-2.5-VL + GPT-4o to generate video descriptions.
* Strong Baselines: Demonstrates that small specialized models (e.g., TiRoBERTa) outperform large LLMs (e.g., GPT-4o by 15% F1).

---

> ### Author Rebuttal · Authors · 2025-07-28
>
> Thank you for your time and the detailed and constructive feedback. We will use the insights to further improve the paper and will clarify the following points.
>
> > **Review:** "Unclear implementation details... The iterative expansion process lacks algorithmic details... Training details for monolingual models (TiROBERTa, TiELECTRA) are omitted."
>
> Thank you for these suggestions. We will revise Section 3.2 to be more explicit with a more detailed description of our iterative sampling algorithm. We used the Gensim library for word2vec, trained on our full 4.1M comment corpus, which inherently captures code-mixed language. We will add a more detailed algorithmic description of our iterative expansion process. Regarding the monolingual models (TiROBERTa/TiELECTRA), their pre-training was part of a prior work we cited, Gaim et al. (2021). For the reader's convenience, we will add a summary of their pre-training corpora and hyperparameters to the appendix and highlight the original reference.
>
> > **Review:** "No discussion of model biases (e.g., performance variance across topics/scripts)."
>
> Regarding bias, we will expand our analysis in Section 5 by discussing the per-class performance disparities (a results Table will be added in the appendix) as a proxy for model bias, noting, for instance, how all models struggle with certain minority topic classes.
>
> > **Review:** "...generated video descriptions are only used for LLMs, not fine-tuned models. Why don't Integrate video descriptions into fine-tuning?"
>
> This is an excellent question. Our rationale was that the generated video descriptions are long-form text, and concatenating them with comments often exceeds the maximum input length (e.g., 256-512 tokens) of the standard encoder models. LLMs, with their much larger context windows, are better suited to digesting this auxiliary information. We agree that exploring sophisticated multimodal fusion techniques for incorporating this rich context into encoder models is a valuable direction for future work, which our dataset now facilitates. We will add this clarification to the paper.
>
> > **Review:** "The single linear head for all 11 labels (across 3 tasks) may limit task-specific learning. Why don't you use three task-specific heads?"
>
> We chose a single output layer as a strong, standard 'hard parameter sharing' baseline. This approach encourages the shared encoder to learn universal features beneficial to all tasks. We agree that comparing different head architectures is an interesting research question and will add this to our future work discussion.

---

> > ### Comment · Reviewer_9xmK · 2025-08-05
> > **Thanks for your response**
> >
> > Thanks for your response. It would be great if you could share any analysis results about model biases during rebuttal.

---

> > ### Author Response · Authors · 2025-08-06
> > **Model Biases Analysis**
> >
> > Thank you for the follow-up. We interpret the *model biases* query as performance variance across different data subsets (e.g., topics, sentiment, and scripts), which is crucial to investigate. Here are some of our key findings:
> >
> > **Topic and Sentiment Disparities:**
> > Our class-level analysis of the fine-tuned encoder models reveals significant performance disparities. Models perform well on the majority classes like `political` topics (up to 71.21% F1) and `negative` sentiment (up to 81.32% F1), but struggle with minority classes like `sexist` topics and `neutral` sentiment (as low as 1.53% F1). This reflects both natural data distribution and the inherent difficulty of nuanced content classification.
> >
> > Importantly, our results show that ***multi-task joint learning helps mitigate these disparities*** to a substantial degree. For instance, TiRoBERTa's F1 score on the challenging `sexist` class improves from 31.78% to 46.30% with joint learning, demonstrating a direct benefit of our proposed approach.
> >
> > **LLM Performance Variance:**
> > The generative LLMs exhibit even more pronounced disparities. Claude Sonnet 3.7 shows highly asymmetric zero-shot performance (44.18% F1 on `abusive` vs. 72.58% on `not abusive`), despite our test set being balanced for the `abusiveness` task (comprised of 50% samples from each class). Most striking is LLaMA-3.2's extreme bias, classifying 96.9% of comments as `abusive`, yielding only 6.36% F1 on `not abusive` content in zero-shot settings. Our qualitative assessment confirms that this heavily stems from poor Tigrinya comprehension and inefficient tokenization, with a 2.31 token-to-character ratio compared to only 0.20 for English (lower is better). All model predictions are available in the public GitHub repository linked in the paper.
> >
> > **Script-Based Robustness:**
> > Our initial analysis indicates that by training on mixed-script data (both Ge'ez and Romanized), the models become more robust across test samples using both writing systems. While these qualitative results are encouraging, a full quantitative evaluation of script-specific performance is planned for the camera-ready version to provide a definitive analysis.
> >
> > ---
> > We believe these analyses provide valuable insights into the performance variance of models on the new TiALD benchmark. The full result tables and detailed analysis will be included in the camera-ready version. To substantiate these claims in the interim, an early version of these results is available in the public arXiv version of our paper (appendix, page 16, https://arxiv.org/pdf/2505.12116), which we will expand upon in the final submission.
> > Thank you again for your constructive engagement.

---

### Official Review · Reviewer_3CEd · 2025-07-01

**Rating:** 5
**Confidence:** 4

**Summary:**

This paper introduces TiALD (Tigrinya Abusive Language Detection), a multi-task benchmark dataset comprising 13,717 YouTube comments annotated by native speakers for abusiveness detection, sentiment analysis, and topic classification. The work addresses content moderation for Tigrinya, a low-resource language with  approx. 10 million speakers. The authors propose an iterative term clustering approach for data selection and establish baselines using fine-tuned models and LLMs, demonstrating that specialized small models outperform frontier LLMs in this low-resource setting.

**Dataset Code Accessibility:**

Yes

**Dataset Code Comments:**

Yes the dataset is readily accessible on huggingface and is well documented and in a usable format.

**Ethical Comments:**

The paper addresses ethical concerns adequately with IRB approval and proper consent procedures.

**Ethical Considerations:**

No, there are no or only very minor ethics concerns

**Final Justification:**

While the missing granular annotations limits the use of the dataset, the joint annotations do mitigate the impact to some extent. I would like to recommend acceptance conditional on the joint annotation be explicitly added and clarified in the final camera ready version.

**Limitations Weaknesses:**

- The authors acknowledge focusing only on explicit forms of abuse, missing implicit toxicity, microaggressions, and subtle prejudice. This is a significant limitation for real-world content moderation applications where implicit harm is prevalent.
- The binary abusiveness classification lacks granularity compared to recent work that distinguishes between hate speech, harassment, offensive language, and other forms of abuse. This limits the dataset's utility for nuanced content moderation systems.

**Strengths Contributions:**

- This work addresses a critical gap in NLP resources for African languages. The creation of a comprehensive, multi-task annotated dataset for Tigrinya is valuable for the research community and represents substantial effort in data collection and annotation.
- The iterative seed-expansion sampling strategy using word embeddings is well-motivated for low-resource settings. This approach achieves better lexical diversity (27.6% type-to-token ratio) compared to keyword-based sampling (18.2%), addressing the challenge of creating balanced datasets without extensive toxic word lists.
- The dataset's accommodation of both Ge'ez script (70%) and Latin transliterations (30%) reflects real-world usage patterns in Tigrinya social media, making the resource more practically relevant.

---

> ### Author Rebuttal · Authors · 2025-07-28
>
> Thank you for your time and insightful feedback, and for recognizing the value of our contribution to NLP for African languages.
>
> > **Review:** "The binary abusiveness classification lacks granularity compared to recent work that distinguishes between hate speech, harassment, offensive language, and other forms of abuse. This limits the dataset's utility for nuanced content moderation systems."
>
> We agree that a more granular annotation of abusiveness is a crucial direction for future research, as we highlighted in our Limitations section. In this work, our primary goal is to establish a foundational resource with highly reliable labels (indicated by the high IAA, $\kappa=0.758$) that enables more granular future extensions. One such possible route is to hierarchically categorize samples with the `Abusive` label into sub-classes, an approach shown to be effective for an English dataset by Song et al. (2021), a work we cite.
>
> More importantly, our multi-task design already provides a path toward this nuance. The joint annotations allow for the analysis of interactions between `abusiveness` and `topic` tasks. For instance, comments labeled as `Abusive` + `Political` can be interpreted as a form of political hate speech, while those labeled `Abusive` + `Sexist` represent misogynistic attacks. Similar inferences can be made for other topics covered in our TiALD dataset, where `Abusive` + `Racial` or `Abusive` + `Religious` comments can point to racial or religious intolerance. Our dataset directly supports this type of nuanced investigation. While these multi-task label interactions do not completely substitute for direct, fine-grained annotations, they offer a deeper, multi-faceted analysis that is vital for real-world moderation systems.
>
> We will revise the paper to make these points more explicit and emphasize how the combination of our three tasks provides a richer signal than a single abusiveness label alone.
> Thank you once again for highlighting this important discussion.

---

> > ### Comment · Reviewer_3CEd · 2025-08-04
> >
> > While the missing granular annotations limits the use of the dataset, the joint annotations do mitigate the impact to some extent. I would like to recommend acceptance conditional on the joint annotation be explicitly added and clarified in the final camera ready version.

---

> > > ### Author Response · Authors · 2025-08-04
> > >
> > > Thank you for the follow-up and for considering our rebuttal. We appreciate your feedback and confirm that we will clarify how the joint annotations provide a path toward a more nuanced analysis, while also encouraging fine-grained annotations as future work.

---

### Official Review · Reviewer_KLVy · 2025-07-03

**Rating:** 4
**Confidence:** 5

**Summary:**

The work presents a dataset for sensitive content classification name TiALD of a low resource language called Tigrinya. The authors describe the label categorization, data collection, annotation and quality control process to create the dataset and present baseline empirical results of popular encoder models and prompt engineering for GPT-like models.

**Dataset Code Accessibility:**

Yes

**Ethical Considerations:**

No, there are no or only very minor ethics concerns

**Final Justification:**

Raising score from 2-> 4 based on rebuttal and future inclusion of suggested changes in camera ready paper.

**Limitations Weaknesses:**

- Novelty severely restricted and limited to a single low resource language dataset. The data sampling and modeling strategies are common place in industry and academia.
- Scale of the dataset presented limited to < 14k samples.
- Only binary classification results have been presented
- 3 labels in topic classification have < 1k samples indicated in table 1. Unsure how these classes perform as in training or evaluation as per label empirical results not available.
- Precision recall numbers are strongly necessary for labels without enough data to study them in an imbalanced dataset

**Strengths Contributions:**

- New dataset of a low resource language
- Details challenges faced in quality control and data collection process
- comprehensive evaluation using encoder and decoder based language models

---

> ### Author Rebuttal · Authors · 2025-07-28
>
> Thank you for your time and feedback. We would like to clarify several aspects of our work.
>
> > **Review:** "Novelty severely restricted and limited to a single low resource language dataset. Scale of the dataset presented limited to < 14K samples."
>
> We respectfully clarify that our core contribution is the creation of the first large-scale, multi-task benchmark for Tigrinya, a severely under-resourced language spoken by over 10 million people. Creating such foundational resources is a primary goal of the NeurIPS Datasets and Benchmarks track. Our dataset, with 13,717 human-annotated samples, is substantial relative to similar datasets in the literature and one of the largest for an African language. Furthermore, the high inter-annotator agreement represents a meticulous annotation effort by nine native speakers, making it a valuable gold-standard asset for future work. Therefore, the work being on one language is not a limitation but a designed need for a focused work, i.e., a multi-lingual study is not the claimed scope of this research.
>
> > **Review:** "Only binary classification results have been presented."
>
> Our benchmark is inherently multi-task and not limited to binary classification. It features three distinct tasks: binary *Abusiveness*, 4-class *Sentiment*, and 5-class *Topic* classification. This is a core feature of our work, detailed in Section 3.3.
>
> We wish to clarify that, for the encoder-based models, we presented experimental results across all three tasks (Tables 2 and 4). For the generative LLMs, we focused our initial experiments on the primary task of abusiveness detection due to the high computational cost required for multi-task generation with these larger models. We thank the reviewer for pointing this out, and we will include the full results for the sentiment and topic classification tasks for LLMs in the final version of the paper.
>
> > **Review:** "Precision recall numbers are strongly necessary for labels without enough data to study them in an imbalanced dataset"
>
> We agree that detailed per-class metrics are important. In the final version, we will expand the appendix to include tables with Precision, Recall, and per-class F1 scores for all tasks to provide this valuable granularity. The observed class imbalance is a genuine reflection of real-world data, and our benchmark's value lies in capturing and presenting these challenges to the community.
>
> While we are unable to include the detailed evaluation results with this reply due to length, we have performed a similar analysis on the preprint arXiv version of the paper, and we will expand it further in the final submission. Furthermore, we made the dataset, training code, evaluation script, trained models, and the predictions of all baseline models publicly available in the GitHub repository linked in the paper.

---

### Note · Authors · 2025-08-12

We thank all reviewers for their time and feedback. We have addressed the points raised in the reviews, and we will provide additional details in the final submission as discussed. It is encouraging that the reviewers recognized our work's contributions, such as creating a critical resource for a low-resource language, the well-motivated novel sampling strategy, and the strong baselines. For full transparency and reproducibility, we have made our dataset, evaluation code, training code, trained models, and the final predictions of all baseline models publicly available on the GitHub repository linked in the paper. Thank you!

---

### Decision · Program_Chairs · 2025-09-18

**Decision:**

Accept (poster)

**Comment:**

This paper introduces TiALD, a multi-task benchmark dataset for abusive language detection in Tigrinya, a severely under-resourced African language. The dataset consists of 13,717 annotated YouTube comments, covering three tasks: abusiveness detection, sentiment analysis, and topic classification. The authors propose an iterative term clustering approach for data selection and establish baselines using fine-tuned models and LLMs, demonstrating that smaller multi-task models outperform frontier LLMs in this low-resource setting.

Strengths:
* Addresses a critical gap in NLP resources for low-resource and African languages, with high potential impact for online safety and content moderation.
* Multi-task design (abusiveness, sentiment, topic) provides richer annotations than single-label datasets, enabling nuanced downstream research.
* The authors provide comprehensive evaluations.

Weaknesses:
* Some implementation and experimental details are under-specified (e.g., iterative sampling method, model training details, handling of code-mixing).
* Evaluation coverage is incomplete: LLM experiments focus only on abusiveness detection, with limited analysis for sentiment and topic classification. Besides, fine-tuning LLMs has not been explored.
* The abusiveness task lacks fine-grained distinctions (e.g., hate speech vs. harassment), which reduces the dataset’s utility for nuanced moderation.

The authors clarified misunderstandings during the rebuttal, which has addressed most concerns from the reviewers. While the dataset has limitations ( lack of fine-grained labels, incomplete experimental coverage), it provides a valuable and much-needed resource for an underserved language and aligns well with the Datasets & Benchmarks track.